# Reproductive individuality of clonal fish raised in near-identical environments and its link to early-life behavioral individuality

Ulrike Scherer [1,2,3] ✉, Sean M. Ehlman [1,2,3], David Bierbach[1,2,3], Jens Krause[1,2,3] & Max Wolf[1,3]

Recent studies have documented among-individual phenotypic variation that emerges in the absence of apparent genetic and environmental differences, but it remains an open question whether such seemingly stochastic variation has fitness consequences. We perform a life-history experiment with naturally clonal fish, separated directly after birth into near-identical (i.e., highly standardized) environments, quantifying 2522 offspring from 152 broods over 280 days. We find that (i) individuals differ consistently in the size of offspring and broods produced over consecutive broods, (ii) these differences are observed even when controlling for trade-offs between brood size, offspring size and reproductive onset, indicating individual differences in life-history productivity and (iii) early-life behavioral individuality in activity and feeding patterns, with among-individual differences in feeding being predictive of growth, and consequently offspring size. Thus, our study provides experimental evidence that even when minimizing genetic and environmental differences, systematic individual differences in life-history measures and ultimately fitness can emerge.

Phenotypic differences among individuals from the same species abound throughout the animal kingdom[1–3], with substantial consequences for fitness, ecology and evolution[4–8]. It is commonly thought that such individuality is caused by genetic and/or environmental differences. More recently, evidence is accumulating that even among individuals with near-identical genetic and environmental backgrounds substantial phenotypic differences can arise[9–11]: isogenic fruit flies, for example, develop differences in locomotor handedness and wing-folding, phototaxis, and object-fixated locomotion[12–14]; naturally clonal fish separated directly after birth into highly standardized environments develop repeatable differences in activity levels that are already present from the first day of life[15,16] (see[17,18] for related findings in inbred mice and clonal pea aphids).

Such findings are important as they demonstrate that (i) our understanding of genetic and environmental variation and the way they combine to generate phenotypic variation is incomplete and (ii) even minute genetic and/or environmental differences can have profound consequences for phenotypic variation[19–22]. Up to now, however, all recent studies on the emergence of variation in the absence of apparent genetic and environmental differences have focused on characterizing emergent behavioral differences (and their neurobiological underpinnings). It remains an open question whether these differences can be best understood as minor and inconsequential 'noise' or 'idiosyncrasies', or whether and to what extent these differences really matter.

One of the most direct ways to answer this question is to investigate whether phenotypic differences that emerge under genetic and environmental standardization extend to those aspects of the phenotype that directly affect fitness. The goal of the present study is to do exactly this. We performed an experimental study with a live-bearing,

¹SCIoI Excellence Cluster, Technische Universität Berlin, 10587 Berlin, Germany. ²Faculty of Life Sciences, Humboldt-Universität zu Berlin, 10117 Berlin, Germany. ³Department of Fish Biology, Fisheries, and Aquaculture, Leibniz Institute of Freshwater Ecology and Inland Fisheries, 12587 Berlin, Germany. ✉e-mail: u.k.scherer@gmail.com

naturally clonal freshwater fish, the Amazon molly, *Poecilia formosa*. Directly after birth, we separated genetically identical individuals ($N = 34$) into near-identical environments and reared them under highly standardized conditions for 280 days (i.e., 40 weeks). We utilized automated high-resolution tracking of activity and feeding patterns to characterize early-life behavioral profiles over the first 28 days of their lives (daily recordings for 10 h at 0.2 s resolution, amounting to a total of 9520 recording hours and 171.4 million data points)[23]. We then characterized reproductive profiles: recording the onset of reproduction, the size of each brood produced (i.e., the number of offspring per brood; in total, $N = 152$ broods), and the size of all offspring produced ($N = 2522$ offspring), thereby observing $4.5 \pm 1.1$ broods (mean ± SD) per female (gestation takes ~30 days[24,25]). Individual body size was measured weekly.

We focus on three key research questions. First, do genetically identical individuals separated at birth into highly standardized environments develop significant among-individual differences in reproductive traits, i.e., repeatable differences in offspring size and/or brood size? Second, are among-individual differences in reproductive traits, if present, indicative of differences in (i) life-history productivity, i.e., the ability to produce new biomass[26,27] and/or (ii) how individuals balance the trade-off between brood size vs. offspring size [28-31]? Third, are reproductive differences, if present, related to early-life behavioral differences? Such a link could be possible via growth, where behavior (in particular feeding behavior) could be linked to growth and growth, in turn, to reproductive output[28,32,33]. As an underlying assumption, and as shown previously[15,16], we expected repeatable early-life behavioral variation. Whenever appropriate, we focus on repeatability as a key parameter to quantify and test for individuality[3,34,35].

## Results
### Early-life behavioral individuality
We find that genetically identical individuals separated into near-identical environments on the day they were born exhibit strong behavioral individuality during the first 4 weeks of life, both in activity (repeatability ($R$) for activity = 0.371, 95% CI = [0.329, 0.413]) and feeding behavior ($R = 0.183$, 95% CI = [0.145, 0.224]). These repeatable differences even increase when controlling for within-individual variation caused by individuals growing and becoming older during our observations (adjusted $R$ for activity = 0.571, 95% CI = [0.532, 0.621]; adjusted $R$ for feeding = 0.238, 95% CI = [0.194, 0.285]). Daily activity and feeding behavior are negatively correlated (estimate = −9.134, 95% CI [−10.550, −7.719], $p$-value < 0.001, $R^2 = 0.164$, Supplementary Table 7); this may be explained by more active fish also being more active during the feeding period, thereby having less time to feed at the stationary food resource.

### Reproductive individuality and life-history productivity variation
We find consistent among-individual differences in both the average size of offspring (Fig. 1a, $R = 0.396$, 95% CI [0.308, 0.484]) and the number of offspring produced (Fig. 1b, $R = 0.177$, 95% CI = [0.117, 0.238]) over consecutive broods. Interestingly, the trade-off between brood size and offspring size explains only a small amount of the variation observed (estimate = −0.013, 95% CI [−0.018, 0.007], $p$-value < 0.001, partial $R^2 = 0.115$) (Fig. 1c, Supplementary Table 8). Even when controlling for this trade-off, among-individual differences remain: given the same brood size, onset of reproduction, and size at parturition, some individuals consistently produce larger offspring than others (adjusted $R$ for offspring size = 0.134, 95% CI = [0.085, 0.192]); similarly, when controlling for onset of reproduction, size at parturition, and offspring size, some individuals consistently produce larger broods than others (adjusted $R$ for brood size = 0.077, 95% CI = [0.050, 0.114]). These findings strongly suggest that – next to developing repeatable among-individual differences in offspring size

and brood size – these genetically identical individuals, raised individually in highly standardized environments, also differ in life-history productivity.

We stress that both offspring and brood size are among the most direct fitness components one can measure, and seemingly small – but repeatable – differences in these traits may have profound long-term consequences. This can be seen, for example, when considering the cumulative number of offspring produced, where even relatively minor individual differences in brood size, when expressed consistently, result in large among-individual differences in total reproductive output (Fig. 1d).

All analyses on female reproductive output are controlled for descent (i.e., mother ID) and female size at parturition (see Supplementary Note 5 'Effect of female size on reproductive output' for an illustration of the effect of female size at parturition on offspring and brood size and Supplementary Table 9 for statistical analyses). Moreover, we controlled experimentally for potential effects of breeding tanks and/or males (see 'Reproductive profiles' in Methods). Statistical analyses confirming that our results are robust towards potential male/ tank IDs are provided in Supplementary Note 3 'Robustness of results with respect to potential male and/or tank effects'.

### Link between behavioral and reproductive individuality
Despite no direct link between our two behavioral measures, activity and feeding, and our three reproductive traits, i.e., offspring size (Fig. 2a, b), brood size (Fig. 2c, d), and onset of reproduction (Fig. 2e, f) (Supplementary Table 9), we find an indirect link between one of our behavioral traits and reproduction: fish that spend more time feeding grow to a larger size (estimate = 0.007, 95% CI [0.003, 0.012], $p$-value = 0.002, partial $R^2 = 0.266$; Fig. 3b and d), and larger fish, in turn, produce larger offspring (Fig. 3c; estimate = 0.619, 95% CI [0.299, 0.939], $p$-value < 0.001, partial $R^2 = 0.129$). Larger fish also start reproducing later (estimate = 55.537, 95% CI [25.189, 85.886], $p$-value = 0.001, partial $R^2 = 0.310$; Fig. 3f), but there is no effect on brood size (Fig. 3e) (Supplementary Table 11).

In contrast to feeding behavior, we find no indirect link between activity and reproduction (Fig. 3a, Supplementary Table 10), nor do we find an effect of our behavioral traits on individual growth rates (predicted from fitted growth curves; see Methods) (Supplementary Table 10). We note growth rate and onset of reproduction are positively correlated: fish that grow faster start reproducing later (estimate = 980.383, 95% CI [235.137, 1725.629], $p$-value = 0.012, partial $R^2 = 0.188$), but there is no effect of growth rate on brood and offspring size (Supplementary Table 11). Throughout, qualitatively the same results are obtained when statistically controlling for potential effects of breeding tanks and/or males (see Supplementary Note 3 'Robustness of results with respect to potential male and/or tank effects').

## Discussion
We find that genetically identical individuals, raised separately and in highly standardized environments, develop repeatable differences in key reproductive characteristics. In particular, when considering consecutive broods, individuals differ consistently in how many offspring they produce and in how large these offspring are. While we find evidence for a weak trade-off between offspring size and number, repeatable among-individual differences are observed even when controlling for this trade-off, as well as for body size and onset of reproduction, providing clear evidence that individuals differ in life-history productivity. While previous studies have provided firm evidence that substantial among-individual variation in anatomical-, behavioral- and neurobiological traits can emerge even in the absence of apparent genetic and environmental differences[10,13,15], the current study builds on and substantially extends these studies by demonstrating that the emerging variation extends to aspects of the phenotype that are directly associated with fitness.

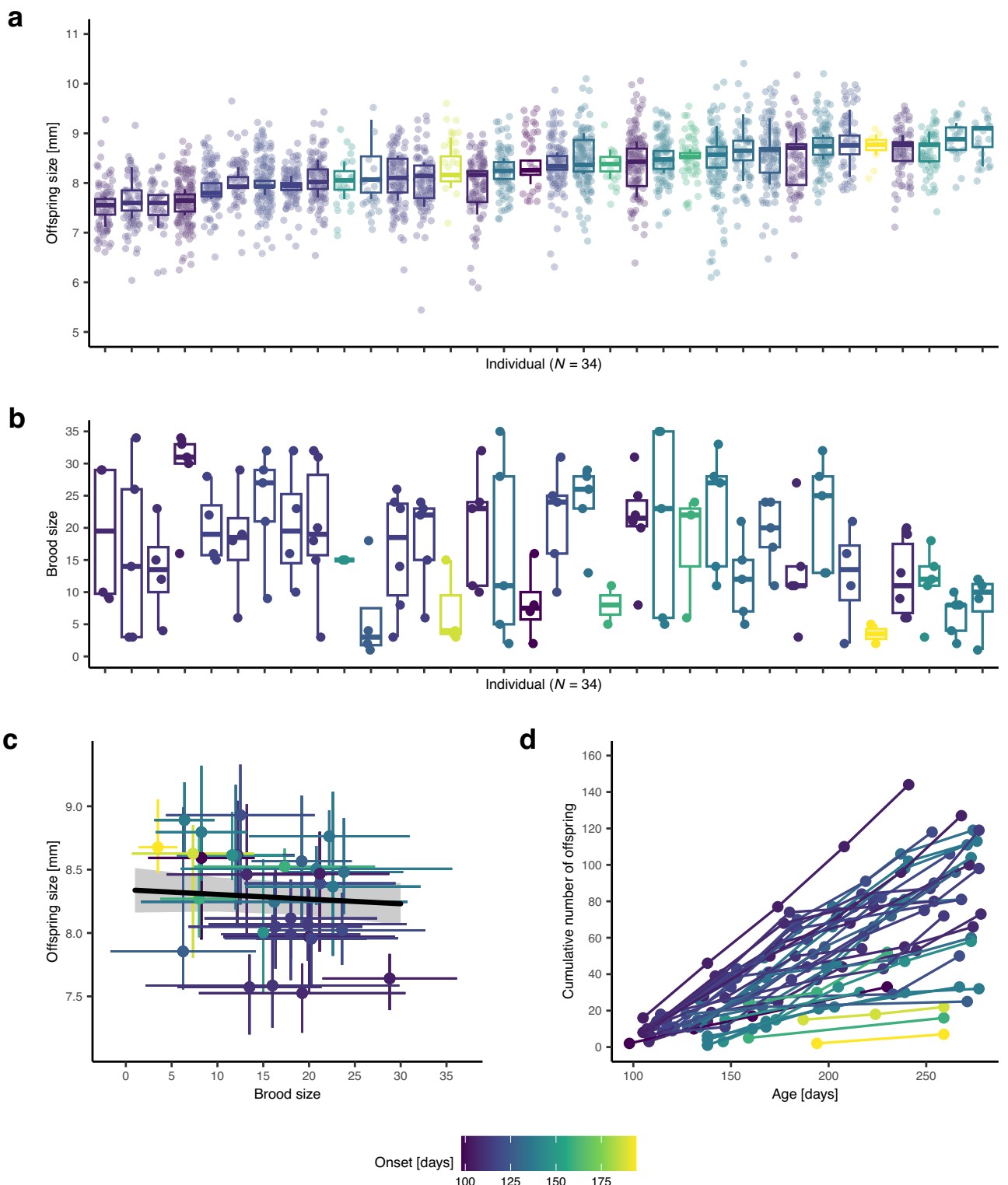

**Fig. 1 | Reproductive individuality. a**, **b** Individuals (*N* = 34) differ consistently in the size and number of offspring produced over successive broods (we recorded 152 broods and measured *N* = 2522 offspring from 144 broods, no size measurements for 8 broods); boxes are sorted by median offspring size; shown are median (middle line), 25ᵗʰ to 75ᵗʰ percentile (box), and 5ᵗʰ to 95ᵗʰ percentile (whiskers) as well as the raw data (points) for each individual. **c** The brood size vs. offspring size trade-off explains only very little of the variation; shown are individual means (points) ± SD (error bars) in brood/offspring size (*N* = 2522 offspring from 144 broods). The regression line (black) and 95% confidence interval (gray shadow) were estimated via linear mixed-effects model. **d** Differences in reproductive productivity can have profound long-term consequences because reproductive output accumulates over time; shown is the cumulative number of offspring produced by individuals over the first 280 days of life. **a**–**d** Coloration by onset (yellow represents late onset, purple represents early onset).

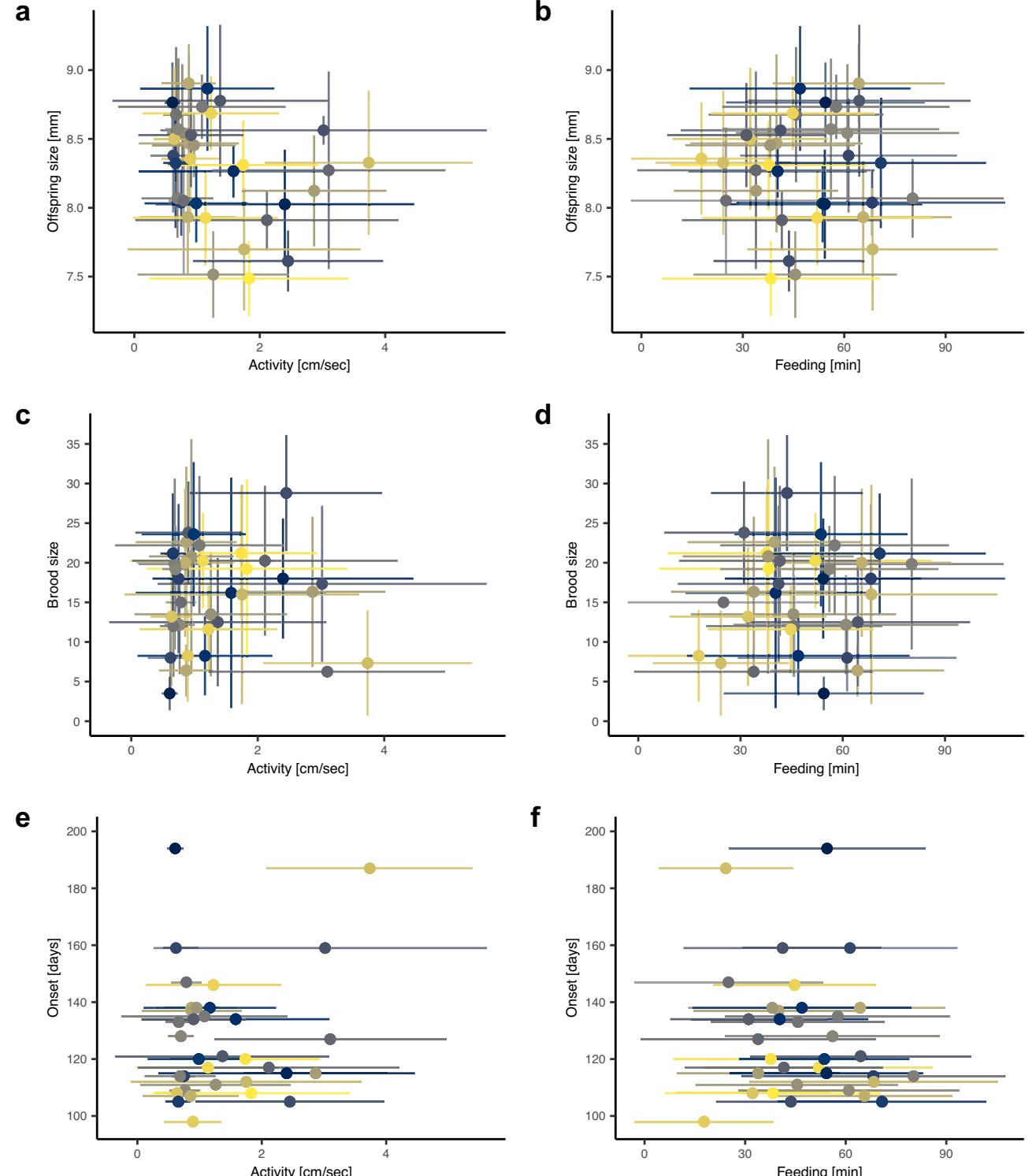

**Fig. 2 | No direct link between early-life behavioral individuality and reproductive individuality.** We find no effect of early-life behavior (activity and feeding) on offspring size (**a**, **b**), brood size (**c**, **d**), or onset of reproduction. **e**, **f** Shown are individual means (points, *N* = 34 individuals) ± SD (error bars) (*N* (activity measurements) = 941, *N* (feeding measurements) = 932, *N* (size measured offspring) = 2522, *N* (broods) = 152). **a**–**f** Individuals are colored differently.

Interestingly, we find no direct link between early-life behavioral differences and differences in reproductive traits. It is conceivable, however, that a direct link between early-life behavioral and later-in-life reproductive traits may only become apparent in non-benign and/or more complex environments. To give a concrete example, in our experimental set-up, there are only minimal differential costs and benefits associated with different behavioral phenotypes: fish were presented with a stationary food resource, located at a standardized position in the tanks with no additional structures. There was no need to search for food and no cost associated with exploiting the stationary food resource. In contrast, in a more naturalistic context, activity might be linked to the ability to find food and the exploitation of a food resource might be risky[36,37]. That said, it will be interesting to see whether future studies, taking the above and other factors into

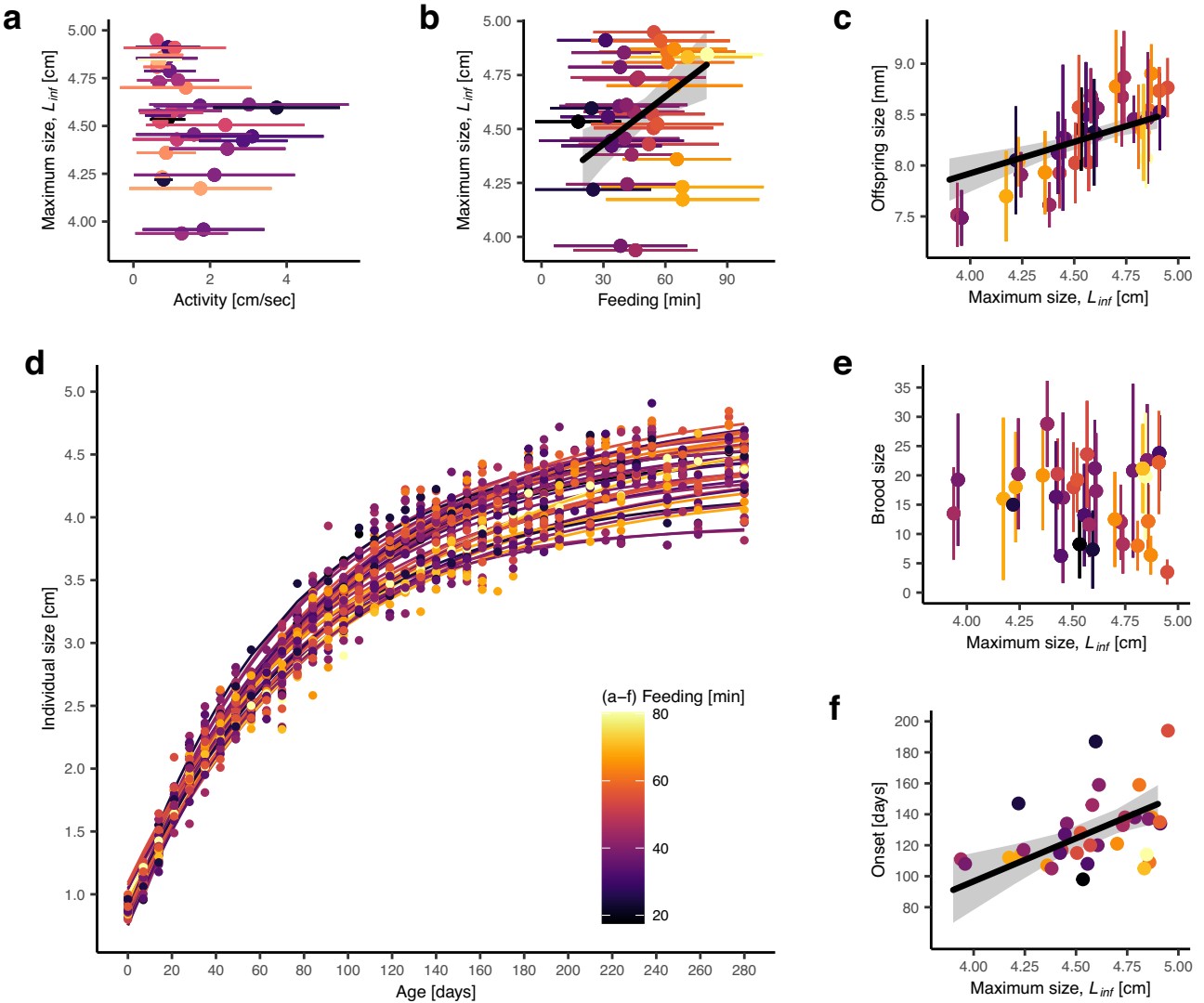

**Fig. 3 | Indirect link between early-life differences in feeding behavior and offspring size. a** There is no effect of early-life activity on the maximum predicted size $L_{inf}$, but early-life feeding behavior and reproductive output are indirectly connected via growth: (**b**) individuals that feed more grow to a larger size (feeding is presented as mean ± SD, $N$ = 932 feeding measurements), and (**c**) larger fish produce larger offspring. **f** Larger fish also start reproducing later but (**e**) individual maximum size and brood size are not linked. **a–e** Shown are individual means

(points) ± SD (error bars) ($N$ (activity measurements) = 941, $N$ (feeding measurements) = 932, $N$ (size measured offspring) = 2522, $N$ (broods) = 152). **b**, **c**, **f** Regression lines (black) and 95% confidence intervals (gray shadow) were estimated via linear mixed-effects models. **d** Shown are individual von Bertalanffy growth curves (lines) and raw data (points). **a–f** Coloration by average time spent feeding (yellow represents long feeding times, black represents short feeding times).

account, will be able to establish a link between early-life behavioral individuality and reproduction.

While we do not find a direct link between reproduction and early-life behavior, we find the size of offspring produced to be indirectly linked to early-life feeding behavior: some individuals feed consistently more than others, individuals that feed more grow to a larger predicted final size, and individuals with a larger final size have larger offspring (but not larger broods). The allocation of resources into increasing offspring size, rather than offspring number, may represent an adaptation to environments in which larger offspring have higher fitness, e.g., resource limitation, competition, or high juvenile mortality (cannibalism or size-dependent predation) (reviewed in ref. 29). All of the above factors may apply to our experimental design: during the reproduction period, individuals were fed a standardized amount of food that they shared with a *P. mexicana* male, which we kept in the female's tank as a sperm donor, potentially causing both resource limitation and competition. Furthermore, offspring were removed from the females' tanks directly after parturition, which decreased the

risk of cannibalism but may have caused perceived predation from the female's perspective.

We find that systematic (i.e., repeatable) among-individual differences in key fitness components can emerge even in the absence of apparent genetic and environmental differences. Future work may compare the observed differences in repeatedly expressed reproductive traits to other iteroparous species, taking genetic and/or environmental variation into account, which will give us a better understanding of the magnitude of the observed differences. Furthermore, we will need to investigate both the causes and consequences of the observed differences. First, all individuals in our study were exposed to one (very specific) environment, and it will be important to investigate whether the observed differences can also be detected in other environments. It will be particularly informative to include predation risk, a key determinant of fitness and major factor shaping life-history trade-offs[38–42], and to investigate, for example, if high-productivity individuals are differently affected by predators than low-productivity individuals. Second, future studies should investigate

the mechanistic causes underlying the development of systematic differences in reproductive traits in the absence of apparent genetic and environmental differences. In principle, the same set of factors (stochastic developmental processes, minute genetic and/or environmental differences in combination with positive feedback loops, and pre-birth influences including epigenetic and maternal effects) that are thought to explain the emergence of behavioral individuality in the absence of apparent genetic and environmental differences may also give rise to reproductive individuality[10,16,19–22]. It is intriguing to speculate that pre-birth processes (e.g., within-brood variation in maternal egg provisioning or in the exact starting time of embryonic development) initiate the observed reproductive differences, e.g., by creating early-life physiological differences. It is also interesting to note that mechanisms such as developmental instability and stochasticity may themselves be selected for as a means of generating adaptive variation through bet-hedging[43]. Detailed studies are needed to further investigate these and other hypotheses relating to underlying mechanisms driving individuality. Third, in order to evaluate the consequences of the observed differences in brood and offspring size, it will be important to investigate whether and to what extent these differences are heritable. Do offspring from mothers with larger offspring (or larger brood sizes) sire larger offspring (or large brood sizes) themselves? While all our individuals are genetically identical, such inheritance is still conceivable, for example via epigenetic mechanisms[44–46].

The study of among-individual phenotypic variation is one of the central themes in ecology and evolution. While such variation is thought to be caused by genetic and environmental differences, evidence is accumulating that even when minimizing genetic and environmental differences, substantial among-individual variation can emerge. But does such seemingly stochastic variation really matter? Here, we show that the emergent among-individual variation extends to aspects of the phenotype that directly affect fitness. Put differently, we find that among-individual variation that arises under highly standardized conditions reflects more than just 'idiosyncrasies' or 'noise' – it really matters.

## Methods

All animal care and experimental protocols complied with local and federal laws and guidelines and were approved by the appropriate governing body in Berlin, Germany, the Landesamt fur Gesundheit und Soziales (LaGeSo G-0224/20).

### Study species and holding conditions

Amazon mollies used in our experiment were obtained from a stock kept at Humboldt-Universität zu Berlin (Berlin, Germany). The lineage used in our experiment was established prior to the start of the experiment by holding a single Amazon molly from the stock population as well as a *P. mexicana* male and their resulting offspring in a 50-liter tank (~20–50 fish, 12:12 h light:dark cycle, air temperature control at ~24 ± 1 °C, weekly water changes, fish were fed with Sera vipan baby powder food twice a day). Before the experiment, we separated potential mothers (i.e., large females, all of whom were sisters) from the isolated lineage and let them give birth in individual tanks; this allowed us to track the mother ID ($N = 3$) of individuals used in our experiment.

### Early-life behavior

Test fish were transferred to individual observation tanks on the day they were born (see Supplementary Fig. 1 for an illustration of experimental tanks and Supplementary Table 1 for statistical analyses confirming environmental standardization). Behavioral observations started the next day, i.e., the first full day of life. We recorded individuals daily over the first 28 days of their life from above with a Basler acA5472 camera (5 frames per second). Activity was observed over the first 8 h of each day, followed by a 2-h feeding observation period.

During the feeding period, individuals were presented with a 'food patch', that was positioned at a standardized location in the tank (Supplementary Fig. 1). Food patches were prepared every 2–3 days using Sera vipan baby powder food and agar (protocol is provided in Supplementary Note 2 'Food patch preparation').

In total, we collected 952 recordings of daily activity and feeding behavior, respectively (34 individuals, each individual recorded over 28 days). To assess activity, we used a total of 941 recordings (mean ± SD recording length: 471.7 ± 29.9 min; 11 days were removed from the data due to technical issues). To calculate how much time individuals spent feeding, we processed 932 feeding recordings (mean ± SD recording length = 120.2 ± 11.2 min; 20 recordings were removed due to technical issues). Recordings were tracked using the software Biotracker[47], and the movement data obtained from Biotracker 3.2.1 (csv-files with xy-coordinates over time) were processed (visualization, calculation of metrics) with a custom-made repository we developed for this purpose[48]. Individual activity was calculated from xy-coordinates over time, in steps of 0.2 s, as the average distance moved (cm) in 1 s. Individual time spent feeding was calculated as the amount of time an individual spent in the 'feeding zone', a 5 × 13 cm large zone surrounding the food patch[47,48].

### Reproductive profiles

After our early-life behavioral observations (including the above described 28-day behavioral observations as well as a 9-day behavioral assay phase that was performed as part of a separate project, for more details see Supplementary Note 7), individuals were transferred to individual breeding tanks (11 liters each, visual separation between individual tanks, equipped with a small plastic pipe (length = 4 cm, diameter = 2 cm) and 'sera biofibres', a loose bundle of green plastic fibers, resembling thread algae). A unified flow-through water system ensured standardized water conditions and a single *P. mexicana* male was placed in each tank. The Amazon molly is a gynogenetic species, i.e., sperm from one of the parental species (*P. mexicana*, *P. latipinna*) is needed to trigger embryogenesis but the male's DNA is not incorporated into the offspring's genome[49–52] (but see[53] for the rare possibility of paternal introgression). Breeding tanks were checked for offspring daily. Once a female gave birth to a brood, all offspring were photographed and counted. Offspring standard length (i.e., the length from the tip of the snout to the end of the caudal peduncle) was measured from the photos using ImageJ[54]. In total, we recorded 152 broods and measured the size of 2522 offspring from 144 broods (no measurements for 8 broods). Individuals produced on average 4.5 ± 1.1 broods (mean ± SD). We excluded all individuals from our analyses with no or partial reproductive data ($N = 11$ females). Feeding was standardized: twice a day for 5 days a week, fish received 1/64 tsp (up to the age of 70 days) or 1/32 tsp (from the age 70 to 280 days) of powder food.

In order to experimentally control for potential systematic male and/or breeding tank effects on female reproductive output, females were swapped once a week in a randomized manner between the different breeding tanks (while males remained in their tank). Over the course of the experiment, females visited 22 ± 2 males and 19 ± 1 breeding tanks (mean ± SD; the number of males exceeds the number of tanks, as some males were replaced, see Supplementary Note 3 'Robustness of results with respect to potential male and/or tank effects'). By providing each female with a series of different tanks (and thus males), we ensured that potential differences between males and/or tanks did not cause systematic differences in female reproductive characteristics (i.e., repeatable differences in brood size and offspring size). We chose to swap females between breeding tanks rather than males because swapping males would have made it impossible to distinguish between the potential effects of breeding tanks and female identity. As a result of our experimental approach, we cannot isolate the effects of breeding tanks from the effects of males,

but we were able to control for both factors simultaneously. See Supplementary Note 3 'Robustness of results with respect to potential male and/or tank effects' for further analyses demonstrating that all our results and conclusions are indeed robust with respect to potential variation caused by breeding tanks and/or males.

### Long-term growth

Standard lengths of focal individuals were measured from photos once a week, using ImageJ[54] ($34 \pm 1$ measurements per individual, mean $\pm$ SD). We fit individual growth curves using the von Bertalanffy growth model[55], a logistic function commonly used to model fish growth. Estimated parameters in this function are the theoretical age when size is zero ($t_O$), the growth coefficient ($K$), and the maximum predicted (i.e., asymptotic) size ($L_{inf}$). For all analyses, we used predicted sizes estimated using individual growth curves rather than raw measurements and characterized individual growth via $K$ and $L_{inf}$ obtained from those individual growth curves.

### Statistical analysis

**General details.** Data were analyzed in R version 4.2.1[56]. Most parsimonious LMs (linear models) and LMMs (linear mixed-effect models; models built using the lme4-package[57]) were fit via stepwise-backward removal of non-significant predictors. Model assumptions were visually assured using residual- and q-q plots. In the main text, we report estimates and $p$-values for significant predictors only. The effects of covariates are presented in Supplementary Note 4 'Model output main analyses'. Most importantly, individual size at parturition is related to both the number and size of offspring produced (model summaries in Supplementary Table 9, illustration in Supplementary Fig. 2); and we found mother ID ($N = 3$) to be related to individual growth and reproductive output (Supplementary Table 9 and 10). We therefore included mother ID as a covariate in all models (LMs and LMMs), and individual size at parturition wherever appropriate. In all LMMs, test fish ID was included as a random term. Individual size on the first day of life did not affect early-life behavior, reproductive output, or growth (Supplementary Tables 12-14 in Supplementary Note 6 'No effects of size at birth on behavior, reproduction, and growth'), and was therefore not considered during analyses. For significant predictors, we calculated partial $R^2$ using the sensemakr- and r2glmm-package[58,59] (for LMs and LMMs, respectively). Complete model summaries of all full (containing all predictors) and final models (containing significant predictors only) are provided in Supplementary Note 4 'Model output main analyses'. Model summary tables (including marginal and conditional $R^2$ following[60]) were built using the package sjPlot[61].

**Repeatabilities.** We estimated repeatabilities with 95% CIs (confidence intervals) in two ways: first by building LMMs with only the target variable as response and female ID as random term, but no predictors (i.e., raw repeatability), and then by adding fixed effects to the model, allowing us to estimate the amount of variation caused by consistent between-individual differences while controlling for variation explained by other factors (i.e., adjusted repeatability). The significance of consistent among-individual differences was derived from the 95% CI being distinctly different to 0 (95% CI based on 1000 model simulations)[62].

To test for 'reproductive individuality', we first calculated raw repeatabilities in the size of broods ($N = 152$ broods) and the size of offspring ($N = 144$ broods, average offspring size per brood) over all broods produced. Then, to test for among-individual differences in productivity, we adjusted repeatabilities of both brood and offspring size for onset, female size at parturition, and mother ID. For the model on brood size, we additionally included offspring size as predictor and vice versa (i.e., we accounted for brood size, offspring size, and onset trade-offs).

To test for early-life behavioral individuality, we calculated raw repeatabilities for activity ($N = 941$ observations of 34 individuals) and time spent feeding (LMM with $N = 932$ observations of 34 individuals), observed daily over the first 28 days of life. We also adjusted repeatabilities for size and age (week 1–4, categorical variable), i.e., we accounted for variation that was caused by individuals growing and aging over the observation period. We further included a size-age interaction term as predictor (activity was differently affected by size, depending on age). For repeatability calculations, activity was log-transformed for normality. We tested if early-life activity (response) and feeding (predictor) are correlated (LMM $N = 931$ observations of 34 individuals).

**Link between early-life behavior and reproduction.** To test for a direct behavior-reproduction link, we built three models with either brood size (LMM with $N = 152$ broods), offspring size (LMM with $N = 144$ broods), or reproductive onset (LM with $N = 34$ individuals) as response. In all models, activity and feeding (averaged over 28 days) were modeled as predictors.

To test for an indirect behavior-reproduction link, we first tested for a link between behavior and growth: we fit one model on the predicted final size $L_{inf}$ (LM with $N = 34$ individuals) and growth rate $K$ (LM with $N = 34$ individuals), with activity and feeding as predictors (average behavior over 28 days). When having $K$ as the response, we additionally included $L_{inf}$ as a covariate to control for the effect $L_{inf}$ has on $K$; i.e., bigger fish grow slower to their final size $L_{inf}$, (LM with $L_{inf}$ as response and $K$ as well as mother ID as predictors: intercept [CI] = 6.09 [5.8, 6.4]; estimate of $K$ [CI] = −16.9 [−20.0, −13.8], $R^2 = 0.89$). To test for the link between growth and reproduction, we fit a model on each brood size (LMM with $N = 152$ broods), offspring size (LMM with $N = 144$ broods), and onset (LM with $N = 34$ individuals); including $K$ and $L_{inf}$ as predictors and female age at parturition (not in the onset-model) as a covariate.

### Reporting summary

Further information on research design is available in the Nature Portfolio Reporting Summary linked to this article.

## Data availability

The data generated in this study have been deposited in Figshare (https://doi.org/10.6084/m9.figshare.23971599.v1)[63]. Source data are provided with this paper.

## Code availability

The code used to calculate daily activity and feeding behavior is publicly available on GitHub (https://github.com/UlrikeScherer/Fish-Tracking-Visualization)[48].

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

## Acknowledgements

We thank Luka Stärk, David Strasiewsky, Serafina Wersing, Olivia O'Connor, Phoebe Schladitz, Fernando Bernstein, Anton Heyder, Mira Turi, and Nicolas Gheorghiu for assistance with animal care and husbandry. This work was supported by the Deutsche Forschungsgemeinschaft (DFG) under Germany's Excellence Strategy – EXC 2002/1 'Science of Intelligence' – project number 390523135 (U.S., S.M.E., D.B., J.K., M.W.).

## Author contributions

J.K. and M.W. acquired funding. U.S., S.M.E., D.B., J.K. and M.W. conceived and designed the experiment. U.S. conducted the experiment. U.S., S.M.E., D.B. and M.W. outlined the data analysis. U.S. conducted the data analysis. U.S. and M.W. wrote the first manuscript draft. U.S., S.M.E., D.B., J.K. and M.W. commented on the manuscript and substantively contributed to the final version.

## Funding

## Competing interests

The authors declare no competing interests.
