## [Peer Review File · Nature Communications]

Reproductive individuality of clonal fish raised in near-identical environments and its link to early-life behavioral individualityReviewers' Comments:

Reviewer #1:

Remarks to the Author:

The paper by Scherer et al reports the results of a study designed to determine whether individual differences in behavior in early life among genetically identical individuals is related to life history and fitness traits in adulthood. The authors intensively measured the activity and foraging behavior of genetically identical *P. formosa* in early life and then monitored their reproduction. They found that there were consistent individual differences in behavior, growth rate, and life history traits. Early life foraging behavior was predictive of eventual offspring size via its effects on growth.

Overall this line of research on the causes and consequences of individual differences in behavior in the *P. formosa* system is very powerful and this study makes an important contribution by showing that this variation has relevance to fitness.

This is a compact, efficient paper and a well designed study. Whether the variation reported here has relevance to natural populations remains to be seen. Does demonstrating that it exists warrant publication in *Nat Communications*?

Overall the methods and analyses seem sound, but I do take some issue with the way that the authors frame the study. Several places in the MS (e.g. line 68, 198, 219) state that the animals were reared in the "absence" of environmental differences. However the authors themselves acknowledge that there is probably microenvironmental/experiential variation (line 66) that the animals experience, despite the experimenters' best efforts to keep the environments as similar as possible. Given the evidence for maternal effects in this study (line 297; there were 3 moms), clearly differences in the maternal environment also matters. The authors should dial back on their claims about the "absence" of environmental variation. There is clearly important un-measured or unidentified environmental variation, which is different from the "absence" of environmental variation.

Here are more minor comments:

The authors should consider how some of the phenotypic variation might reflect measurement error

Lines 103-110, 121-124: these repeatabilities are pretty low. The authors should discuss how/whether they think these small effect sizes are likely to be meaningful in natural environments

Line 108: I would have predicted that increased activity would be positively correlated with feeding, if animals have to move/be active in order to get food. Can you explain?

Line 114: indicate in the results how many of these traits are affected by female body size, and influenced by maternal ID

Line 156: how is this a "twin" study? Did it compare mono- and dizygotic twins reared in the same versus different environments?

Line 266: was there an effect of paternal ID, i.e. paternal effects?

Reviewer #2:

Remarks to the Author:

In the "Reproductive individuality of clonal fish raised in identical environments and its link to early-life behavioral individuality", Scherer et al. performed a tightly controlled life history experiment to investigate the question, if among individual differences in the Amazon molly that are neither derived by genes nor the environment have fitness consequences.

Strengths: The authors provide compelling evidence that 34 Amazon mollies show individual differences in their fitness-related properties: Offspring size and Brood size (Fig. 1a-b).

Major criticism: While the authors can provide direct evidence for fitness-related individual differences in the Amazon molly, I have three major areas of concern. The first two have to do with the prerequisites of the study, and the third is about the mechanistic inside of the study.

The prerequisites of the study are that all the variation observed is independent of genetic or environmental differences, as mentioned for example in "despite being genetically identical and being raised in identical environments" (lines 37-38).

While I do not doubt that the Amazon molly is a fantastic model and its ameiotic parthenogenesis is a unique vertebrate reproduction mode, a genetic identity assumption requires careful referencing and a potential source of error analysis. Warren et al., 2018 describe genetic variability in Amazon mollies derived from different locations and possible introgression of paternal genomic elements. Paternal introgression makes using many different males, as done in this study, a potential source of some genetic variability. The potential summed effects on genetic identity should be ideally measured or at least very carefully discussed.

A similar problem is the assumption of environmental identity. The description in the supplement speaks for a well-standardized environment. Still, it is unclear how this would differ from any other well-controlled experimental setup that would justify the term identical. Potential error sources are differences between arenas in the center and the periphery and shared light sources etc. Given the importance of environmental identity for this study, this should be carefully disclosed, and the term identical potentially mellowed down. Similar concerns could be raised about the food that could be variable between batches which might be even to a certain degree measurable. As the authors pointed out that genetic and environmental identity is central to the study, extra care should be taken, and if possible experimental evidence should be provided to strengthen the fact that all measures have been taken to make the environment as identical as possible.

The last major concern is the mechanistic inside of the study. The study demonstrates direct fitness-related individual differences in the Amazon molly. It provides some indirect evidence to link these to feeding preferences in early life, but, although probably beyond the scope of this paper, even the discussion lacks a mechanistic explanation of how these individual differences arise. Are these differences that originate in the nervous system or physiological differences? What is the stochastic developmental process relevant to these fitness differences?

Minor :

Why are the 34 experimental animals from three maternal fish? Are these three direct descendants from a single mother?

Line 53: experimental

Reviewer #1 (Remarks to the Author):

We are grateful to the reviewer for the thoughtful and constructive comments, which greatly contributed to the improvement of our manuscript.

The paper by Scherer et al reports the results of a study designed to determine whether individual differences in behavior in early life among genetically identical individuals is related to life history and fitness traits in adulthood. The authors intensively measured the activity and foraging behavior of genetically identical *P. formosa* in early life and then monitored their reproduction. They found that there were consistent individual differences in behavior, growth rate, and life history traits. Early life foraging behavior was predictive of eventual offspring size via its effects on growth.

Overall this line of research on the causes and consequences of individual differences in behavior in the *P. formosa* system is very powerful and this study makes an important contribution by showing that this variation has relevance to fitness.

This is a compact, efficient paper and a well designed study. Whether the variation reported here has relevance to natural populations remains to be seen. Does demonstrating that it exists warrant publication in *Nat Communications*?

Overall the methods and analyses seem sound, but I do take some issue with the way that the authors frame the study. Several places in the MS (e.g. line 68, 198, 219) state that the animals were reared in the “absence” of environmental differences. However the authors themselves acknowledge that there is probably microenvironmental/experiential variation (line 66) that the animals experience, despite the experimenters’ best efforts to keep the environments as similar as possible. Given the evidence for maternal effects in this study (line 297; there were 3 moms), clearly differences in the maternal environment also matters. The authors should dial back on their claims about the “absence” of environmental variation. There is clearly important un-measured or unidentified environmental variation, which is different from the “absence” of environmental variation.

This is an important comment, and we fully agree with the reviewer. In the revised manuscript, we are now rephrasing throughout, i.e., at all places in the manuscript, we have changed “identical environments” to “near-identical environments” or “highly standardized environments”; and “absence of environmental differences” to “absence of apparent environmental differences”.

Here are more minor comments:

The authors should consider how some of the phenotypic variation might reflect measurement error

We thank the reviewer for this important comment. Measurement errors, if occurring in a systematic fashion, can lead to an overestimation of among-individual differences (resulting in an overestimation of repeatability); alternatively, if measurement errors occur in an unsystematic, random fashion, they can lead to an overestimation of within-individual variation (resulting in an underestimation of repeatability).

Systematic measurement errors could be introduced, e.g., by residual environmental variation in our experimental set-up. In response to this comment (and a related comment by Reviewer #2 on potential environmental variation), we have now included substantial additional analyses of potential environmental effects during our early-life behavioral observations (**Supplementary information "1 Behavioral observations"**). More specifically, we tested for potential effects of the tank system (1-4), distance to the filter (0-2m), and centrality (center vs. periphery) on our two early-life behavioral variables (activity and feeding). Importantly, we found that none of the environmental parameters had a significant effect on our behavioral variables (**Supplementary Table S1**).

During breeding, we prevented systematic environmental differences between test individuals by swapping females once a week between breeding tanks and thus males (as males remained in their tank). In response to this comment (and a related comment by Reviewer #2 on potential environmental variation), in the revised manuscript, we are now more explicit regarding our experimental standardization of potential environmental differences, i.e., male/tank effects (**lines 284-300** in the main text). Furthermore, we now provide additional analyses showing that our results and conclusions are robust towards potential male/tank effects (for more details, please see below our answer to the question/comment: "was there an effect of paternal ID, i.e. paternal effects?").

For all behavioral, reproductive, and morphological traits measured, we took great care to keep the effects of potential unsystematic, random measurement errors to a minimum. For example, to reduce measurement error in behavioral traits, we recorded individuals with high resolution over an extensive amount of time (daily recordings for 10 hours at 0.2 resolution for 28 days). To give a second example, in order to reduce potential measurement error in body size, we estimated individual growth curves based on all measurements taken over the course of the experiment; and for a specific date of interest, we then used predicted sizes that were determined based on the entirety of individual measurements rather than a single measurement.

Lines 103-110, 121-124: these repeatabilities are pretty low. The authors should discuss how/whether they think these small effect sizes are likely to be meaningful in natural environments

This is an important comment. We note that the repeatabilities of our two behavioral variables (activity and feeding) are moderately high – considering that, for example, a meta-analysis on the repeatability of behavior¹ reports an average repeatability of 0.37. While the repeatabilities of our two reproductive variables (offspring size and brood size) are lower, these variables are arguably the most direct measures of fitness components one could take, consequently, even smaller between-individual differences can be expected to have important consequences. This is, for example, illustrated in **Figure 1d** (main text), where we see how even small repeatable differences in brood size can produce substantial long-term differences in cumulative reproductive output between individuals. We briefly discuss this important point in our manuscript (**lines 113-118**):

“We stress that both offspring- and brood size are the most direct fitness components one can measure, and seemingly small - but repeatable - differences in these traits may have profound long-term consequences. This can be seen, for example, when considering the cumulative number of offspring produced, where even relatively minor individual differences in brood size, when expressed consistently, result in large among-individual differences in total reproductive output (Figure 1d).”

1. Bell, A. M., Hankison, S. J. & Laskowski, K. L. The repeatability of behaviour: a meta-analysis. *Anim Behav* **77**, 771–783 (2009).

Line 108: I would have predicted that increased activity would be positively correlated with feeding, if animals have to move/be active in order to get food. Can you explain?

This is an interesting question. In our experimental set-up, individuals were fed with a stationary food resource at a fixed location. We may assume that individuals that were more active were generally more active throughout the day – including the feeding period. And fish that spend more time swimming around during the feeding period have less time to feed left. As a response to this comment, we added this explanation to our main text (**lines 91-95**):

“Daily activity and feeding behavior are negatively correlated [...], this may be explained by more active fish also being more active during the feeding period, thereby having less time to feed at the stationary food resource.”

Line 114: indicate in the results how many of these traits are affected by female body size, and influenced by maternal ID

We thank the reviewer for this important comment. Both offspring size and brood size are affected by maternal ID and female size at parturition (**Supplementary Table S9**).

Importantly, all our results are controlled for these effects. As a response to this comment, we have now added this information to the results section in the main text (**lines 119-122**):

*“All analyses on female reproductive output are controlled for descent (i.e., mother ID) and female size at parturition (see **Supplementary Information “5 Effect of female size on reproductive”** for an illustration of the effect of female size at parturition on offspring and brood size and **Supplementary Table S9** for statistical analyses).”*

Line 156: how is this a “twin” study? Did it compare mono- and dizygotic twins reared in the same versus different environments?

In order to avoid ambiguity, we have now deleted this phrase (**line 153**).

Line 266: was there an effect of paternal ID, i.e. paternal effects?

This is an important comment. We note that Reviewer #2 had a related comment (first comment) and we thus felt that it is appropriate to elaborate on this issue more extensively (our responses to both comments closely resemble each other).

We first emphasize that, for our experimental design, we put substantial thought and effort into experimentally controlling for systematic male effects on female reproductive output. In order to do so, we provided each female with a series of different males rather than one specific one, that is, females were swapped between the different breeding tanks once a week in a randomized manner (while males remained in their tank). Thus, in total, over the course of the experiment, each female had access to 22 ± 2 different males (mean \pm SD over all females). This way, we ensured that potential systematic effects of males cannot cause systematic differences in female reproductive characteristics (i.e., repeatable differences in brood size and offspring size). In fact, we believe that this design should give

us a conservative measure of female reproductive individuality, as possible systematic male effects should tend to promote variation within females, thereby underestimating female reproductive individuality. As a response to this comment (and the related comment by Reviewer #2), in the revised version of our manuscript, we now put substantial effort into carefully discussing the possibility of systematic male effects and the way we experimentally controlled for it (**lines 284-300**).

Next, we stress that, a key goal of our experimental design was to simultaneously control for potential effects of males and breeding tanks – thereby allowing us to clearly isolate the effect of female identity on repeatable differences in brood size and offspring size, the focus of our paper. This is why chose to swap females and not males between the different breeding tanks (swapping males would have implied that we cannot distinguish between a potential effect of breeding tank and female ID). As a consequence, with our experimental design, we are not able to distinguish between potential male effects and potential effects of breeding tanks (while controlling for both of them at the same time).

Keeping this in mind, as a further response to this comment (and the related comment by Reviewer #2), in the revised manuscript, we have now included substantial further analyses testing whether our results are robust towards potential male/tank effects (**Supplementary information “3 Robustness of results with respect to potential male and/or tank effects”**). In short, for all broods, we used our data to back-calculate the presumed males that triggered embryonic development/tanks where embryonic development was triggered (in the following: male/tank ID) and repeated all analyses involving female reproductive output, controlling for male/tank ID. Importantly, we find no effect of male/tank ID on female reproductive individuality (i.e., qualitatively the same repeatabilities for our two reproductive traits offspring size and brood size; **Supplementary Table S2**) or its link to early-life behavior (i.e., no direct link between early-life behavioral differences and differences in reproductive traits but an indirect link between feeding and offspring size, mediated by growth; **Supplementary Tables S4 and S5**). Thus, we find that all our results and conclusions are robust, i.e., qualitatively the same when statistically controlling for male/tank ID (additionally to controlling experimentally for potential male/tank effects).

We would like to emphasize that the fact that all our results are robust with respect to the above statistical controls of male/tank effects supports the effectiveness of our experimental design, which was aimed at experimentally controlling for potential effects of males and breeding tanks (**lines 137-141 in Supplementary information**).

We note that the above analysis shows that male/tank ID is associated with repeatable differences in offspring size (but not brood size) (**Supplementary Table S2**). We stress that this finding should be treated with caution, in particular with respect to interpreting it as a

potential male effect, as our experiment was neither designed to detect potential male effects nor to distinguish between potential male effects and potential effects of breeding tanks but rather to detect female effects while controlling for both male and tank effects (main text **lines 284-300**).

Reviewer #2 (Remarks to the Author):

We are grateful to the reviewer for the thoughtful and constructive comments, which greatly contributed to the improvement of our manuscript.

In the "Reproductive individuality of clonal fish raised in identical environments and its link to early-life behavioral individuality", Scherer et al. performed a tightly controlled life history experiment to investigate the question, if among individual differences in the Amazon molly that are neither derived by genes nor the environment have fitness consequences.

Strengths: The authors provide compelling evidence that 34 Amazon mollies show individual differences in their fitness-related properties: Offspring size and Brood size (Fig. 1a-b).

Major criticism: While the authors can provide direct evidence for fitness-related individual differences in the Amazon molly, I have three major areas of concern. The first two have to do with the prerequisites of the study, and the third is about the mechanistic inside of the study.

The prerequisites of the study are that all the variation observed is independent of genetic or environmental differences, as mentioned for example in "despite being genetically identical and being raised in identical environments" (lines 37-38).

While I do not doubt that the Amazon molly is a fantastic model and its ameiotic parthenogenesis is a unique vertebrate reproduction mode, a genetic identity assumption requires careful referencing and a potential source of error analysis. Warren et al., 2018 describe genetic variability in Amazon mollies derived from different locations and possible introgression of paternal genomic elements. Paternal introgression makes using many different males, as done in this study, a potential source of some genetic variability. The potential summed effects on genetic identity should be ideally measured or at least very carefully discussed.

This is an important comment. We note that Reviewer #1 had a related comment and that the responses to both comments closely resemble each other.

We first emphasize that, for our experimental design, we put substantial thought and effort into experimentally controlling for systematic male effects on female reproductive output. In order to do so, we provided each female with a series of different males rather than one specific one, that is, females were swapped between the different breeding tanks once a week in a randomized manner (while males remained in their tank). Thus, in total, over the

course of the experiment, each female had access to 22 ± 2 different males (mean \pm SD over all females). This way, we ensured that potential systematic effects of males cannot cause systematic differences in female reproductive characteristics (i.e., repeatable differences in brood size and offspring size). In fact, we believe that this design should give us a conservative measure of female reproductive individuality, as possible systematic male effects should tend to promote variation within females, thereby underestimating female reproductive individuality. As a first response to this comment (and the related comment by Reviewer #2), in the revised version of our manuscript, we now put substantial effort to carefully discuss this issue, that is, the possibility of systematic male effects (including a reference to Warren et al., 2018; **lines 272-276** in the main text) and the way we experimentally controlled for it (**lines 284-300**).

Next, we stress that, a key goal of our experimental design was to simultaneously control for potential effects of males and breeding tanks – thereby allowing us to clearly isolate the effect of female identity on repeatable differences in brood size and offspring size, the focus of our paper. This is why chose to swap females and not males between the different breeding tanks (swapping males would have implied that we cannot distinguish between a potential effect of breeding tank and female ID). As a consequence, with our experimental design, we are not able to distinguish between potential male effects and potential effects of breeding tanks (while controlling for both of them at the same time) (see main text, **lines 284-300**).

Keeping this in mind, as a further response to this comment (and the related comment by Reviewer #2), in the revised manuscript, we have now included substantial further analyses testing whether our results are robust towards potential male/tank effects (**Supplementary information “3 Robustness of results with respect to potential male and/or tank effects”**). In order to do so, for all broods, using three alternative approaches, we used our data to back-calculate the presumed males that triggered embryonic development/tanks where embryonic development was triggered (in the following: male/tank ID). We then repeated all analyses involving female reproductive output, controlling for male/tank ID. Importantly, we find that there is no effect of male/tank ID on female reproductive individuality (in **Supplementary Table S2**) or its link to early-life behavior (**Supplementary Tables S4 and S5**), i.e., we find that all our results are robust and qualitatively the same when controlling statistically for potential male/tank effects (additionally to having experimentally controlled for those effects). To be specific, independent of the approach used to assign male/tank ID, we find:

- qualitatively the same repeatabilities for our two reproductive traits offspring size and brood size (**Supplementary Table S2**).

- as previously, no direct link between early-life behavioral differences and differences in reproductive traits but an indirect link between feeding and offspring size, mediated by growth (**Supplementary Tables S4 and S5**).

We would like to emphasize that the fact that all our results are robust with respect to the above statistical controls of male/tank effects supports the effectiveness of our experimental design, which was aimed at experimentally controlling for potential effects of males and breeding tanks (**lines 137-141 in Supplementary information**).

We note that the above analysis shows that male/tank ID is associated with repeatable differences in offspring size (but not brood size) (**Supplementary Table S2**). We stress that this finding should be treated with caution, in particular with respect to interpreting it as a potential male effect, as our experiment was neither designed to detect potential male effects nor to distinguish between potential male effects and potential effects of breeding tanks but rather to detect female effects while controlling for both male and tank effects (main text **lines 284-300**).

A similar problem is the assumption of environmental identity. The description in the supplement speaks for a well-standardized environment. Still, it is unclear how this would differ from any other well-controlled experimental setup that would justify the term identical. Potential error sources are differences between arenas in the center and the periphery and shared light sources etc. Given the importance of environmental identity for this study, this should be carefully disclosed, and the term identical potentially mellowed down. Similar concerns could be raised about the food that could be variable between batches which might be even to a certain degree measurable. As the authors pointed out that genetic and environmental identity is central to the study, extra care should be taken, and if possible experimental evidence should be provided to strengthen the fact that all measures have been taken to make the environment as identical as possible.

We thank the reviewer for this important comment. In the revised manuscript, we have now included substantial additional analyses on potential environmental effects during early-life behavioral observations (**Supplementary information "1 Behavioral observations"**) as well as during breeding (**Supplementary information "3 Robustness of results with respect to potential male and/or tank effects"**).

Regarding potential environmental variation during the early-life behavioral observations, we have now tested for potential effects of the tank system (1-4), distance to the external filter unit (0-2m), and centrality (center vs. periphery) on our two early-life behavioral

variables (activity and feeding). We find that none of the environmental parameters had a significant effect on our behavioral variables (**Supplementary Table S1**).

Regarding potential environmental variation during breeding, as discussed above (see our response to previous comment), we put substantial thought and effort to experimentally control for that. Most importantly, throughout the breeding phase, females were swapped once a week in a randomized manner between the different breeding tanks (while males remained in their tank), providing each female with a series of approx. 20 different breeding tanks (and thus males). In response to this comment, in the revised manuscript, we now put substantial effort to carefully discuss how we experimentally controlled for potential environmental effects (main text **lines 284-300**). Additionally, in the revised manuscript, we have now included substantial further analyses in order to test whether our results are robust towards potential variation between breeding tanks and/or males (**Supplementary information "3 Robustness of results with respect to potential male and/or tank effects"**). Importantly, we find that all our results are robust, i.e., qualitatively the same when controlling statistically for potential tank/male effects (additionally to having experimentally controlled for those effects) – showing that our experimental design, which was aimed at experimentally controlling for tank/male effects, was successful. For more detailed information please see our response to the previous comment.

As a further response to this comment, we have added detailed information on how observation tanks were illuminated (**Supplementary information lines 48-51**):

*"Observation tanks (**Supplementary Figure S1**) were illuminated individually from below with 4 LEDs per tank (each LED is 100cm, 12V, color temperature = 5500 K, light output = approx. 1570 lumen; tanks were manufactured from white polyethylene, which allowed light from underneath to get through)."*

Potential sources of variation in food quality could be imprecise measurements of ingredients or deviations in cooking or cooling down periods. That said, we took all measures to make food quality as identical as possible, using a fine scale for all measurements and highly standardizing cooking and cooling down periods (**Supplementary information "2 Food patch preparation"**).

We fully agree, despite our best efforts to control and standardize experimental conditions, we will never be able achieve 100% identical environmental conditions for all test individuals. Thus, following the suggestions of the reviewer (and a related comment from Reviewer #1), in the revised manuscript, we are now framing our study more cautiously: "identical environments" is now being rephrased throughout to "near-identical environments" or "highly standardized environments"; and "absence of environmental

differences" is now being rephrased throughout to "absence of apparent environmental differences".

The last major concern is the mechanistic inside of the study. The study demonstrates direct fitness-related individual differences in the Amazon molly. It provides some indirect evidence to link these to feeding preferences in early life, but, although probably beyond the scope of this paper, even the discussion lacks a mechanistic explanation of how these individual differences arise. Are these differences that originate in the nervous system or physiological differences? What is the stochastic developmental process relevant to these fitness differences?

We thank the reviewer for these intriguing questions and we fully agree that, while indeed beyond the scope of this paper, these important issues have to be discussed in our manuscript. In response, in the revised manuscript, we now provide such a discussion (**lines 204-218**):

"Second, future studies should investigate the mechanistic causes underlying the development of systematic differences in reproductive traits in the absence of genetic and apparent environmental differences. In principle, the same set of factors (stochastic developmental processes, minute environmental differences in combination with positive feedback loops, and pre-birth influences including epigenetic and maternal effects) that are thought to explain the emergence of behavioral individuality in the absence of genetic and apparent environmental differences may also give rise to reproductive individuality^{16,19,20,21}. It is intriguing to speculate that pre-birth processes (e.g., within-brood variation in maternal egg provisioning or in the exact starting time of embryonic development) initiate the observed reproductive differences, e.g., by creating early-life physiological differences. It is also interesting to note that mechanisms such as developmental instability and stochasticity may themselves be selected for as a means of generating adaptive variation through bet-hedging⁴³. Detailed studies are needed to further investigate these and other hypotheses relating to underlying mechanisms driving individuality."

Minor :

Why are the 34 experimental animals from three maternal fish? Are these three direct descendants from a single mother?

These are important questions. We chose to use three instead of one mother in order to avoid oddity effects (i.e., reducing the risk of finding results that are specific to one female).

The three mothers used are direct descendants from a single ancestor mother. In the revised manuscript, we are now more specific about the descent (**lines 241-248**).

Line 53: experimental

Using the term “experiential differences” we were referring to differences in the experiences individuals make. Avoiding ambiguity, we have now deleted this term and only write “environmental differences” (**line 35**).

Reviewers' Comments:

Reviewer #1:

Remarks to the Author:

The authors have done a good job responding to the comments from the reviewers. The explanations make sense and the paper has improved.

What I find most interesting about this study is that it illustrates just how incomplete is our understanding of how complex phenotypes arise. Even after going to great lengths to control for genetic and environmental variation, there was still substantial inter-individual variation in behavior, and in fitness. This study underscores just how little we understand about the developmental mechanisms leading to individual phenotypic variation.

I have just one subtle comment/suggestion for the authors to consider. They did a great job changing some of the language to tone down the assumption about their ability to completely control genetic and environmental variation, but there are parts of the introduction (e.g. lines 45-52) that imply that genetic and environmental variation was completely controlled.

For example line 45-6 states, "such findings are important as they demonstrate that genetic and environmental differences are not the only potent source of variation among individuals". I don't think the results of the growing number of deep phenotyping studies on genetically identical individuals calls G and E into question. Instead, they illustrate that we don't know all the sources of G or E variation, such as those listed on line 47, e.g. "pre birth processes (including epigenetics), development per se and/or minor environmental differences". Those are all arguably environmental or genetic influences, they are just hard to measure, often unknown and not easy to identify.

Some of the current language in the paper gives the impression that this study is informing the reader about another source of variation besides G and E that we didn't know about. Instead the study is showing just how much phenotypic variation is still evident even when bending over backwards to control for genetic and environmental sources of variation. That means our understanding of genetic and environmental variation and the way they combine to generate phenotypic variation is clearly incomplete, not that there is another source of variation besides genetic and environmental that we don't know about.

Reviewer #2:

Remarks to the Author:

Reviewer #2 (Remarks to the Author):

We are grateful to the reviewer for the thoughtful and constructive comments, which greatly contributed to the improvement of our manuscript.

In the "Reproductive individuality of clonal fish raised in identical environments and its link to early-life behavioral individuality", **Scherer et al. performed a tightly controlled life history experiment to investigate the question, if among individual differences in the Amazon molly that are neither derived by genes nor the environment have fitness consequences.**

Strengths: The authors provide compelling evidence that 34 Amazon mollies show individual differences in their fitness-related properties: Offspring size and Brood size (Fig. 1a-b).

Major criticism: While the authors can provide direct evidence for fitness-related individual differences in the Amazon molly, I have three major areas of concern. The first two have to do with the prerequisites of the study, and the third is about the mechanistic inside of the study.

The prerequisites of the study are that all the variation observed is independent of genetic or environmental differences, as mentioned for example in "despite being genetically

identical and being raised in identical environments” (lines 37-38).

While I do not doubt that the Amazon molly is a fantastic model and its ameiotic parthenogenesis is a unique vertebrate reproduction mode, a genetic identity assumption requires careful referencing and a potential source of error analysis. Warren et al., 2018 describe genetic variability in Amazon mollies derived from different locations and possible introgression of paternal genomic elements. Paternal introgression makes using many different males, as done in this study, a potential source of some genetic variability. The potential summed effects on genetic identity should be ideally measured or at least very carefully discussed.

This is an important comment. We note that Reviewer #1 had a related comment and that the responses to both comments closely resemble each other.

We first emphasize that, for our experimental design, we put substantial thought and effort into experimentally controlling for systematic male effects on female reproductive output. In order to do so, we provided each female with a series of different males rather than one specific one, that is, females were swapped between the different breeding tanks once a week in a randomized manner (while males remained in their tank). Thus, in total, over the course of the experiment, each female had access to 22 ± 2 different males (mean \pm SD over all females).

Very important, and it was not obvious to me in the previous version!

This way, we ensured that potential systematic effects of males cannot cause systematic differences in female reproductive characteristics (i.e., repeatable differences in brood size and offspring size). In fact, we believe that this design should give us a conservative measure of female reproductive individuality, as possible systematic male effects should tend to promote variation within females, thereby underestimating female reproductive individuality. As a first response to this comment (and the related comment by Reviewer #2), in the revised version of our manuscript, we now put substantial effort to carefully discuss this issue, that is, the possibility of systematic male effects (including a reference to Warren et al., 2018; lines 272-276 in the main text) and the way we experimentally controlled for it (lines 284-300).

Next, we stress that, a key goal of our experimental design was to simultaneously control for potential effects of males and breeding tanks – thereby allowing us to clearly isolate the effect of female identity on repeatable differences in brood size and offspring size, the focus of our paper. This is why chose to swap females and not males between the different breeding tanks (swapping males would have implied that we cannot distinguish between a potential effect of breeding tank and female ID). As a consequence, with our experimental design, we are not able to distinguish between potential male effects and potential effects of breeding tanks (while controlling for both of them at the same time) (see main text, lines 284-300).

Keeping this in mind, as a further response to this comment (and the related comment by Reviewer #2), in the revised manuscript, we have now included substantial further analyses testing whether our results are robust towards potential male/tank effects (Supplementary information “3 Robustness of results with respect to potential male and/or tank effects”). In order to do so, for all broods, using three alternative approaches, we used our data to back-calculate the presumed males that triggered embryonic development/tanks where embryonic development was triggered (in the following: male/tank ID). We then repeated all analyses involving female reproductive output, controlling for male/tank ID. Importantly, we find that there is no effect of male/tank ID on female reproductive individuality (in Supplementary Table S2) or its link to early-life behavior (Supplementary Tables S4 and S5), i.e., we find that all our results are robust and qualitatively the same when controlling statistically for potential male/tank effects (additionally to having experimentally controlled for those effects). To be specific, independent of the approach used to assign male/tank ID, we find:

- qualitatively the same repeatabilities for our two reproductive traits offspring size and brood size (Supplementary Table S2).

- as previously, no direct link between early-life behavioral differences and differences in reproductive traits but an indirect link between feeding and offspring size, mediated by growth (Supplementary Tables S4 and S5).

We would like to emphasize that the fact that all our results are robust with respect to the above statistical controls of male/tank effects supports the effectiveness of our experimental design, which was aimed at experimentally controlling for potential effects of males and breeding tanks (lines 137-141 in Supplementary information).

We note that the above analysis shows that male/tank ID is associated with repeatable differences in offspring size (but not brood size) (Supplementary Table S2). We stress that this finding should be treated with caution, in particular with respect to interpreting it as a potential male effect, as our experiment was neither designed to detect potential male effects nor to distinguish between potential male effects and potential effects of breeding tanks but rather to detect female effects while controlling for both male and tank effects (main text lines 284-300).

This is a very important addition and clarification of the paper that significantly improves it.

A similar problem is the assumption of environmental identity. The description in the supplement speaks for a well-standardized environment. Still, it is unclear how this would differ from any other well-controlled experimental setup that would justify the term identical. Potential error sources are differences between arenas in the center and the periphery and shared light sources etc. Given the importance of environmental identity for this study, this should be carefully disclosed, and the term identical potentially mellowed down. Similar concerns could be raised about the food that could be variable between batches which might be even to a certain degree measurable. As the authors pointed out that genetic and environmental identity is central to the study, extra care should be taken, and if possible experimental evidence should be provided to strengthen the fact that all measures have been taken to make the environment as identical as possible.

We thank the reviewer for this important comment. In the revised manuscript, we have now included substantial additional analyses on potential environmental effects during early-life behavioral observations (Supplementary information "1 Behavioral observations") as well as during breeding (Supplementary information "3 Robustness of results with respect to potential male and/or tank effects").

Regarding potential environmental variation during the early-life behavioral observations, we have now tested for potential effects of the tank system (1-4), distance to the external filter unit (0-2m), and centrality (center vs. periphery) on our two early-life behavioral variables (activity and feeding). We find that none of the environmental parameters had a significant effect on our behavioral variables (Supplementary Table S1).

Regarding potential environmental variation during breeding, as discussed above (see our response to previous comment), we put substantial thought and effort to experimentally control for that. Most importantly, throughout the breeding phase, females were swapped once a week in a randomized manner between the different breeding tanks (while males remained in their tank), providing each female with a series of approx. 20 different breeding tanks (and thus males). In response to this comment, in the revised manuscript, we now put substantial effort to carefully discuss how we experimentally controlled for potential environmental effects (main text lines 284-300). Additionally, in the revised manuscript, we have now included substantial further analyses in order to test whether our results are robust towards potential variation between breeding tanks and/or males (Supplementary information "3 Robustness of results with respect to potential male and/or tank effects"). Importantly, we find that all our results are robust, i.e., qualitatively the same when controlling statistically for potential tank/male effects (additionally to having experimentally controlled for those effects) – showing that our experimental design, which was aimed at

experimentally controlling for tank/male effects, was successful. For more detailed information please see our response to the previous comment.

As a further response to this comment, we have added detailed information on how observation tanks were illuminated (Supplementary information lines 48-51):

“Observation tanks (Supplementary Figure S1) were illuminated individually from below with 4 LEDs per tank (each LED is 100cm, 12V, color temperature = 5500 K, light output = approx. 1570 lumen; tanks were manufactured from white polyethylene, which allowed light from underneath to get through).”

Potential sources of variation in food quality could be imprecise measurements of ingredients or deviations in cooking or cooling down periods. That said, we took all measures to make food quality as identical as possible, using a fine scale for all measurements and highly standardizing cooking and cooling down periods (Supplementary information “2 Food patch preparation”).

We fully agree, despite our best efforts to control and standardize experimental conditions, we will never be able achieve 100% identical environmental conditions for all test individuals. Thus, following the suggestions of the reviewer (and a related comment from Reviewer #1), in the revised manuscript, we are now framing our study more cautiously: “identical environments” is now being rephrased throughout to “near-identical environments” or “highly standardized environments”; and “absence of environmental differences” is now being rephrased throughout to “absence of apparent environmental differences”.

Again, these are very important additions and clarifications. Together with the now much more careful phrasing, they fully resolve my concerns.

The last major concern is the mechanistic inside of the study. The study demonstrates direct fitness-related individual differences in the Amazon molly. It provides some indirect evidence to link these to feeding preferences in early life, but, although probably beyond the scope of this paper, even the discussion lacks a mechanistic explanation of how these individual differences arise. Are these differences that originate in the nervous system or physiological differences? What is the stochastic developmental process relevant to these fitness differences?

We thank the reviewer for these intriguing questions and we fully agree that, while indeed beyond the scope of this paper, these important issues have to be discussed in our manuscript. In response, in the revised manuscript, we now provide such a discussion (lines 204-218):

“Second, future studies should investigate the mechanistic causes underlying the development of systematic differences in reproductive traits in the absence of genetic and apparent environmental differences. In principle, the same set of factors (stochastic developmental processes, minute environmental differences in combination with positive feedback loops, and pre-birth influences including epigenetic and maternal effects) that are thought to explain the emergence of behavioral individuality in the absence of genetic and apparent environmental differences may also give rise to reproductive individuality 16,19, 20, 21. It is intriguing to speculate that pre-birth processes (e.g., within-brood variation in maternal egg provisioning or in the exact starting time of embryonic development) initiate the observed reproductive differences, e.g., by creating early-life physiological differences. It is also interesting to note that mechanisms such as developmental instability and stochasticity may themselves be selected for as a means of generating adaptive variation through bet-hedging⁴³. Detailed studies are needed to further investigate these and other hypotheses relating to underlying mechanisms driving individuality.”

I think this is again an important addition opening the door for future exciting studies.

Minor :

Why are the 34 experimental animals from three maternal fish? Are these three direct descendants from a single mother?

These are important questions. We chose to use three instead of one mother in order to avoid oddity effects (i.e., reducing the risk of finding results that are specific to one female). The three mothers used are direct descendants from a single ancestor mother. In the revised manuscript, we are now more specific about the descent (lines 241-248).

Fine.

Line 53: experimental

Using the term "experiential differences" we were referring to differences in the experiences individuals make. Avoiding ambiguity, we have now deleted this term and only write "environmental differences" (line 35).

I agree that this keeps it simpler.

In summary, I think the revised version of the manuscript by Scherer et al. has substantially improved and has resolved all the major concerns I had with the previous version. I endorse publication in Nature Communications.

REVIEWERS' COMMENTS

Reviewer #1 (Remarks to the Author)

The authors have done a good job responding to the comments from the reviewers. The explanations make sense and the paper has improved.

We sincerely appreciate the positive feedback on our revised manuscript, and we are grateful for the thoughtful comments and insights.

What I find most interesting about this study is that it illustrates just how incomplete is our understanding of how complex phenotypes arise. Even after going to great lengths to control for genetic and environmental variation, there was still substantial inter-individual variation in behavior, and in fitness. This study underscores just how little we understand about the developmental mechanisms leading to individual phenotypic variation.

I have just one subtle comment/suggestion for the authors to consider. They did a great job changing some of the language to tone down the assumption about their ability to completely control genetic and environmental variation, but there are parts of the introduction (e.g. lines 45-52) that imply that genetic and environmental variation was completely controlled.

For example line 45-6 states, "such findings are important as they demonstrate that genetic and environmental differences are not the only potent source of variation among individuals". I don't think the results of the growing number of deep phenotyping studies on genetically identical individuals calls G and E into question. Instead, they illustrate that we don't know all the sources of G or E variation, such as those listed on line 47, e.g. "pre birth processes (including epigenetics), development per se and/or minor environmental differences". Those are all arguably environmental or genetic influences, they are just hard to measure, often unknown and not easy to identify.

Some of the current language in the paper gives the impression that this study is informing the reader about another source of variation besides G and E that we didn't know about. Instead the study is showing just how much phenotypic variation is still evident even when bending over backwards to control for genetic and environmental sources of variation. That means our understanding of genetic and environmental variation and the way they combine to generate phenotypic variation is clearly incomplete, not that there is another source of variation besides genetic and environmental that we don't know about.

We agree that it is essential to clarify that we do not propose another source of variation besides genes and environment. Avoiding any misinterpretation of our study's findings, in the revised manuscript, we now use the phrasing "**apparent** environmental differences" instead of "environmental differences" at two further incidences in the introduction section where we haven't been clear before:

*"It is commonly thought that such individuality (if not minor and inconsequential 'noise' or 'idiosyncrasies') is primarily caused by genetic and/or **apparent** environmental differences." (lines 18-20)*

*"Such findings are important as they demonstrate that genetic and **apparent** environmental differences are not the only potent source of variation among individuals,..." (lines 30-31).*

Reviewer #2 (Remarks to the Author):

We sincerely appreciate the positive feedback on our revised manuscript, and we are grateful for the thoughtful comments and insights.